# Oncogenic Signalling of PEAK2 Pseudokinase in Colon Cancer

**DOI:** 10.3390/cancers14122981

**Published:** 2022-06-16

**Authors:** Céline Lecointre, Elise Fourgous, Ingrid Montarras, Clément Kerneur, Valérie Simon, Yvan Boublik, Débora Bonenfant, Bruno Robert, Pierre Martineau, Serge Roche

**Affiliations:** 1CRBM, University of Montpellier, CNRS, 34000 Montpellier, France; celine.lecointre1@ac-normandie.fr (C.L.); elise.fourgous@crbm.cnrs.fr (E.F.); ingrid.montarras@evotec.com (I.M.); clement.kerneur@ext.inserm.fr (C.K.); valerie.simon@crbm.cnrs.fr (V.S.); yvan.boublik@crbm.cnrs.fr (Y.B.); 2Equipe labellisée Ligue Contre le Cancer, CRBM, University of Montpellier, CNRS, 34000 Montpellier, France; 3Novartis Institutes for BioMedical Research, CH 4033 Basel, Switzerland; dbonenfant@monterosatx.com; 4IRCM, INSERM, University of Montpellier, 34000 Montpellier, France; bruno.robert@inserm.fr (B.R.); pierre.martineau@inserm.fr (P.M.)

**Keywords:** pseudokinase, oncogene, cell signalling, cell migration, actin cytoskeleton, colorectal cancer, phosphoproteomic

## Abstract

**Simple Summary:**

Catalytically inactive kinases, also named pseudokinases, play important roles in the regulation of cell growth and adhesion. While frequently deregulated in human cancer, their role in tumour development is partially elucidated. Here, we report an important tumour function for the pseudokinase PEAK2 in colorectal cancer (CRC) and propose that PEAK2 upregulation can affect cancer cell adhesive properties through an ABL-dependent mechanism to enable cancer progression. Therefore, targeting PEAK2 oncogenic activity with small tyrosine kinases (TK) inhibitors may be of therapeutic interest in colorectal cancer (CRC).

**Abstract:**

The PEAK family pseudokinases are essential components of tyrosine kinase (TK) pathways that regulate cell growth and adhesion; however, their role in human cancer remains unclear. Here, we report an oncogenic activity of the pseudokinase PEAK2 in colorectal cancer (CRC). Notably, high *PRAG1* expression, which encodes PEAK2, was associated with a bad prognosis in CRC patients. Functionally, PEAK2 depletion reduced CRC cell growth and invasion in vitro, while its overexpression increased these transforming effects. PEAK2 depletion also reduced CRC development in nude mice. Mechanistically, PEAK2 expression induced cellular protein tyrosine phosphorylation, despite its catalytic inactivity. Phosphoproteomic analysis identified regulators of cell adhesion and F-actin dynamics as PEAK2 targets. Additionally, PEAK2 was identified as a novel ABL TK activator. In line with this, PEAK2 expression localized at focal adhesions of CRC cells and induced ABL-dependent formation of actin-rich plasma membrane protrusions filopodia that function to drive cell invasion. Interestingly, all these PEAK2 transforming activities were regulated by its main phosphorylation site, Tyr413, which implicates the SRC oncogene. Thus, our results uncover a protumoural function of PEAK2 in CRC and suggest that its deregulation affects adhesive properties of CRC cells to enable cancer progression.

## 1. Introduction

Colorectal cancer (CRC) remains a leading cause of malignancy-related death worldwide. Most of these cancers are sporadic and under the control of genetic, epigenetic, and environmental factors [1]. The current clinical management of localized CRC involves surgical removal of the tumour, which can be associated adjuvant chemotherapy. Targeted therapies (e.g., EGF and EVGF signalling and immune checkpoint inhibitors) have been incorporated in the therapeutic regimen to treat advanced stage tumours [2]. However, tumour recurrence occurs in about 40% of patients, resulting in poor prognosis with 5-year survival rates of less than 10% [2,3]. CRC have been classified upon their genomic profile to improve therapeutic decision [3,4]. As a result, four consensus molecular subtypes (CMS) have been determined that reflect CRC heterogeneity for about 85% of CRC tumours [3]. These include MSI+ tumours with strong immune infiltration (CMS1), Wnt/beta-catenin proliferative tumours (CMS2) and KRAS mutated and metabolic-deregulated tumours with strong immune exclusion (CMS3), and “mesenchymal” tumours with stromal and innate immune infiltration (CMS4). However, this classification was made from early-stage CRC, which does not completely reflect the situation at the metastatic stage because of the plastic nature of metastatic CRC cells [5]. Metastatic cell behaviour is characterized by aberrant adhesive properties and tumour-initiating capacities (also defined as cancer stems cells), which are under the control of the tumour microenvironment [6]. Therefore, targeting this metastatic process may be of obvious therapeutic interest in advanced CRC.

Protein phosphorylation plays a pivotal role in cell biological events that leads to cell growth, adhesion, and fate decision, and deregulation of kinase–substrate interactions often leads to human cancer [7]. As results, protein kinases have become highly attractive targets in medical oncology. However, a small percentage of kinases has been successfully targeted clinically, probably because the precise mechanisms by which they regulate protein phosphorylation is not fully elucidated. This is despite evidence that many are potential candidates for the development of new therapeutics. Notably, the human kinome contains catalytic inactive kinases that play as important roles as active protein kinases in cancers [8,9]. While the precise role of these pseudokinases in cellular signalling remains unclear, recent structural analyses suggest that they function by docking additional kinases for efficient protein phosphorylation and that some of them, after all, possess active kinase activity, thus implicating an unconventional molecular mechanism of protein phosphorylation [8,9].

Several groups including ours have linked PEAK1_3 pseudokinases to cancer (i.e., PEAK1/Sgk269, PEAK2/Pragmin/SgK223/NACK, and PEAK3/C19orf35) [10,11,12,13,14,15,16,17]. These pseudo-kinases belong to signalling proteins that regulate cell adhesion and proliferation induced by growth and adhesive cues [10,14]. They can also localize in the nucleus to induce gene transcription [18]. Mechanistically, they undergo tyrosine phosphorylation, allowing the recruitment of important effectors for intracellular signalling [14]. For instance, SRC-induced PEAK2 phosphorylation on Y413 triggers binding of SH2-containing protein such as CSK to enable tumour cell elongation and migration [11,19]. Compelling evidence now supports an essential role for these pseudokinases in human cancer [10,14]. PEAK1 and 2 are overexpressed in many epithelial cancers and contribute to their tumour progression [10,14]. Notably, their aberrant tumour expression often leads from epithelial cell to mesenchyme transition, which plays critical roles in cancer cell dissemination and therapeutic resistance [13,20,21,22,23,24]. PEAK3 is overexpressed in acute myeloid leukaemia and promotes leukaemic cell growth [16]. 

Crystallographic studies revealed a unified model on how PEAKs pseudokinases may assemble and regulate oncogenic signalling pathways [11,12,25]. They display a classical protein kinase fold but are catalytically inactive because of an inaccessible nucleotide binding site occluded by an inhibitory triad. Importantly, they contain a Split HElical Dimerization (SHED) domain flanking the pseudokinase, which regulates PEAKs’ self-association (Figure 1B) [11,12,15,16,17]. This SHED module is essential for their scaffolding activity of signalling hubs and their oncogenic activity [12,16,17]. While protumour functions for PEAK1 were reported in CRC [26], the oncogenic role of PEAK2 in this cancer was not clearly established. Here, we investigated the oncogenic role of this pseudokinase in CRC.

## 2. Materiel and Methods

### 2.1. Reagents

Antibodies: anti-PEAK2 and anti-PEAK2 pY413 were described in [11]; anti-ABL was described in [27]; anti-ABL pY412, anti-ERK1/2, anti-ERK1/2 pT202/Y204, anti-AKT, anti-AKT pS473, anti-pTyr clone pY1000 Sepharose bead conjugated (PTM Scan, Cell Signalling Technology, Danvers, MA, USA); anti-Hes1, anti-Actin, anti-AMPK, and anti-AMPK pT172; anti-Myc tag (9B11) were from Cell Signalling Technology; antitubulin (gift from N. Morin, CRBM, Montpellier, France); anti-ATOH1 (EPR6419, Abcam, Cambridge, UK); anti-pY418 SRC (Biosource, Camarillo, California, US), anti-pTyr 4G10 (gift from P. Mangeat, CRBM, Montpellier, France); anti-cst1 (that recognizes SRC, FYN and YES) was described in [28]; antirabbit IgG-HRP and antimouse IgG-HRP (GE Healthcare, Chicago, IL, USA). Kinase inhibitors: Imatinib, dasatinib, bosutinib, lapatinib, PP2 were from Sigma-Aldrich, (St. Louis, MI, USA); nilotinib and CSKi (compound **13**) [29] was a gift from Novartis (Basel, Switzerland) and Bristol-Myers Squibb Company (New York, NY, USA), respectively. Fibronectin human plasma (F0895) and rat tail collagen I (#C3867) were from Sigma. Plasmids: pMX-pS-CESAR retroviral vector expressing human SRC, rat PEAK2 fused to the Myc tag at the C-terminus (PEAK2^Myc^), YF (Y391 in the rat sequence corresponding to Y413 in the human sequence) and KD (K997A in the rat sequence corresponding to the K1024 in the human sequence) mutants, pBABE-ABL, pSGT-ABL, pBABE-PEAK2^Myc^, and pcDNA3-PEAK2^Myc^ were described in [11,22,27,30]. GFP-PEAK2^Myc^ and mCherry-PEAK2^Myc^ constructs were obtained by insertion of PEAK2^Myc^ sequence in pEGFP-N1 and pmCherry N1, respectively. pRETRO-SUPER expressing shRNA used in this study were from TRANSAT (Saint-Priest, France) and described in [22]. Targeting sequences inserted in shRNA constructs were GACACTCGGTAGTCTATAC (control) and GTCACAGGCCAAGATAGAA (PEAK2). siRNAs used in this study: siRNA control (Ctrl) 5′TTCTCCGAACGTGTCACGTTT3′ (Eurofins, Luxemburg), siRNA PEAK2 #1 (Dharmacon, Lafayette, CO, USA, siSGK223 J-025870-05), #2 (Dharmacon siSGK223 J-025870-05), and #3 (Qiagen, Hilden, Germany, siSGK223-2458) and siRNA ABL (Dharmacon smartpool siRNA ABL1 M-040285-00).

### 2.2. CRC Cell Lines and Patient Samples

CRC cell lines (HCT116, HT29, Caco2, SW480, SW620, Lovo, LS174T, DLD1, T84, and Colo205) and HEK293T cell lines were obtained from ATCC (Rockville, MD, USA); C0-115 (a gift from Dr Pierre Roux, CNRS of Montpellier) was described in [31]. SRC-transformed NIH3T3 were described in [32]. Patient-derived CRC and CTC cell lines (a gift from Dr. Pannequin, CNRS of Montpellier) we recently generated from metastatic tumours and circulating CRC cells in blood samples and have been described in [33,34]. These include the CPP25, CPP43, and CPP44 cell lines that have been freshly established from a primary metastatic tumour that express mutated KRAS (or mutated BRAF in the case of CPP44) and the CTC44 and 45 cell lines derived from circulating tumour cells (CTC) that express mutated BRAF from two chemotherapy-naive patients with metastatic CRC (stage IV). Transcriptomic analysis was performed on a cohort of 205 stage III CRC samples described in [35]. Relapse Free Survival (RFS) was defined as the time from surgery to the first recurrence and was censored at 5 years as described [35]. RFS probability was analysed using the Kaplan–Meier method, and differences between survival distributions were assessed using the log-rank and Breslow–Gehan tests.

### 2.3. Cell Cultures, Retroviral Infections, and Transfections

Cells were cultured at 37 °C and 5% CO_2_ in a humidified incubator in Dulbecco’s Modified Eagle’s Medium (DMEM) GlutaMAX (Invitrogen, Waltham, MA, USA) supplemented with 10% foetal calf serum (FCS), 100 U/mL of penicillin and 100 µg/mL of streptomycin. CTC cell lines were cultured in M12 medium containing advanced DMEM-F12 (Gibco), 2 mmol/L of l-glutamine, 100 Unit/mL of penicillin and streptomycin, N2 supplement (Gibco, Waltham, MA, USA), 20 ng/mL of epidermal growth factor (R & D) and 10 ng/mL of fibroblast growth factor-basic (R&D) in ultralow attachment 24-well plates (Corning, Corning, NY, USA) as described in [33]. Retroviral production and cell infection were performed as described in [36]. Stable cell lines were obtained by GFP-fluorescence-activated cell sorting. Transient plasmid transfections in HEK293T cells were performed with the jetPEI reagent (Polyplus-transfection) according to the manufacturer’s instructions. For fibronectin cell stimulation, HEK293T cells were seeded overnight on 6-well plates that were coated or not with fibronectin (1/50 dilution) before plasmid transfection (1d). For siRNA transfection, 2.10^5^ cells were seeded in 6-well plates and transfected with 20 nmol of siRNA and 9 µL of Lipofectamine RNAi Max according to the manufacturer’s protocol (ThermoFisher Scientific, Waltham, MA, USA). siRNA used in this study: scramble siRNA (siMock) 5′TTCTCCGAACGTGTCACGTTT3′ that was used as a negative control (Eurofins) and siRNA Abl was described in [27]. 

### 2.4. Soft Agar Colony Formation, Cell Adhesion and Invasion Assays

Colonies formation assays were performed from 1000 cells per well that were seeded in 12-well plates in 1 mL DMEM containing 10% FCS and 0.33% agar on a layer of 1 mL of the same medium containing 0.7% agar. After 18–21 days, colonies with >50 cells were scored as positive. Cell adhesion assays were performed on collagen I (50 μg/mL) or fibronectin (1/50 dilution) coated coverslips as described in [37]. Cell invasion assay was performed in Fluoroblok invasion chambers (BD Bioscience, San Jose, USA) using 50,000 cells in the presence of 100 µL of 1–1.2 mg/mL Matrigel (BD Bioscience) as in [38]. After 24 h, cells were labelled with Calcein AM (Sigma Aldrich) and invasive cells were photographed using the EVOS FL Cell Imaging System (ThermoFisher Scientific). Quantification of the number of invasive cells per well was done with the Image J software (https://imagej.nih.gov/ij/) (accessed on 14 March 2022). 

### 2.5. RNA Extraction and RT-Quantitative PCR

mRNA was extracted from cell lines and tissue samples using the RNeasy plus mini kit (Qiagen) according to the manufacturer’s instructions. RNA (1 µg) was reverse transcribed with the SuperScript VILO cDNA Synthesis Kit (Invitrogen, Waltham, MA, USA). Quantitative PCR (qPCR) was performed with the SyBR Green Master Mix in a LightCycler 480 (Roche, Basel, Switzerland). Expression levels were normalized with the Tubulin human housekeeping gene. Primers used for qPCR: Tubulin, Forward-5′CCGGACAGTGTGGCAA CCAGATCGG3′, Reverse-5′TGGCCAAAAGGACCTGAGCGAACGG3′; PEAK2, Forward-5′GGCCAGGTATGCACAGGTAAT3′, Reverse-5′ AGATCGTCCGATGGTCCTCTT3′. 

### 2.6. Biochemistry and Phosphoproteomic Analyses

Immunoprecipitation and immunoblotting were performed as described in [36]. Briefly, cells were lysed at 4 °C with lysis buffer (20 mM Hepes pH7.5, 150 mM NaCl, 0.5% Triton X-100, 6 mM β-octylglucoside, 10 µg/mL aprotinin, 20 µM leupeptin, 1 mM NaF, 1 mM DTT and 100 µM sodium orthovanadate). 20–50 µg of whole cell lysates were loaded on SDS-PAGE gels and transferred onto Immobilon membranes (Millipore, Burlington, MA, USA). Detection was performed using the ECL System (Amersham Biosciences, Amersham, UK). Original blots can be found in Appendix A. Quantitative phosphoproteomics was performed as in [38]. Briefly, HEK293T cells transfected with indicated constructs for 40 h were lysed in urea buffer (8 M urea in 200 mM ammonium bicarbonate pH 7.5). Phosphopeptides were purified after tryptic digestion of 20 mg (for cells) or of 35 mg (for mouse tumours) total proteins using the PTMScan^®^ Phospho-Tyrosine Rabbit mAb (P-Tyr-1000) Kit (Cell Signalling Technology), according to manufacturer’s protocol. An additional enrichment step using the IMAC-Select Affinity Gel (Sigma Aldrich) was performed to increase the phosphopeptide enrichment. Purified phosphopeptides were resuspended in 10% formic acid and two technical replicates for each sample were analysed using an EASY-nano LC system (Proxeon Biosystems, Odense, Denmark) coupled online with an LTQ-Orbitrap Elite mass spectrometer (Thermo Scientific, Waltham, MA, USA). Each sample was loaded onto a 15 cm column packed in-house with 3 µM ReproSil-Pur C18 (75 µm inner diameter). Buffer A consisted of H_2_O with 0.1% formic acid and Buffer B of 100% acetonitrile with 0.1% formic acid. Peptides were separated using a gradient from 0% to 24% buffer B for 65 min, from 24% to 40% buffer B for 15 min and from 40% to 80% buffer B for 15 min (a total of 95 min at 250 nL/min). Data were acquired with the ‘Top 15 method’, where every full MS scan was followed by 15 data-dependent scans on the 15 most intense ions from the parent scan. Full scans were performed in the Orbitrap at 120,000 resolution with target values of 1 × 10^6^ ions and 500 ms injection time, while MS/MS ion trap scanning parameters were 1 × 10^4^ ions as target value and 200 ms as maximum accumulation time. Database searches were performed with Mascot Server using the human Uniprot database (version 3.87) (EMBL-EBI, Cambridge, UK). Mass tolerances were set at 10 ppm for the full MS scans and at 0.8 Da for MS/MS. Label-free quantification was performed on duplicate LC-MS runs for each sample using Progenesis LC-MS (Nonlinear Dynamics Software)(Waters, Newcastle, UK). Peptide intensities were normalized across all LC-MS runs and normalized peptide intensities were summed for each unique phosphorylated peptide with a Mascot score exceeding 25. These intensities were then used to calculate the log2 fold change ratios of each unique phosphopeptide or each unique protein. In case of ambiguous phosphorylation site assignments, spectra were manually interpreted to confirm the localization of the phosphorylation site using Scaffold (Proteome software; Portland, USA). Phosphoproteomic data is summarized in Appendix A. Protein-protein interaction networks were performed using String (string-db.org) (accessed on 10 may 2022) from our PEAK2^Myc^ phosphoproteomic (this study) and interactomic data (50 top hits) [11]. 

### 2.7. Cell Imaging

Cells seeded on microscope coverslips were fixed with 4% paraformaldehyde in Triton-X100-BRB80 buffer (0.5% Triton-X100, 80 mM PIPES pH 6.8, 1 mM MgCl2, 1 mM EGTA) containing 5% FCS at 37 °C for 20 min. Cells were incubated with the indicated primary antibodies and the corresponding secondary antibodies labelled with Alexa Fluor dyes (488 or 594 nm) and DAPI diluted in PBS buffer. For F-actin cytoskeletal analysis and PEAK2 cell localization, CRC cells were plated on glass coverslips coated with fibronectin as indicated, transfected with indicated constructs for 48 h and subcellular PEAK2 distribution was analysed after cell fixation (4% paraformaldehyde) by direct fluorescence using confocal microscopy (PEAK2^Myc^-GFP). For filopodia formation analysis, transfected cells were plated at a low density (1:10) for 24 h before fixation. When indicated, cells were treated with DMSO (0.1%) or nilotinib (100 nM) 2 h before fixation. PEAK2^Myc^ constructs and F-actin was visualized by Texas red-conjugated phalloidin (1:200 dilution; Abcam, Cambridge, UK) and anti-Myc antibody (1:1000 dilution; 9B11) respectively, after cell fixation (4% paraformaldehyde) and permeabilization (0.05% TRITON for 10 min at room temperature). After mounting with Pro-Long Antidafe Mountant (ThermoFisher), cells were observed with a DMR A micro-scope and a PL APO 63× oil objective (1.32 NA). Images were captured using a piezo-stepper (E662 LVPTZ amplifier; Servo) and a cooled CCD Micromax camera (1300 × 1030 pixels, RS; Princeton Instruments Inc., Trenton USA)) driven by MetaMorph (v 4.5; Universal Imaging Corp., Hialeah, FL, USA). Images were processed with ImageJ (https://imagej.nih.gov/ij/ accessed on 14 March 2022) and used for cell morphological analyses (>50 cells per condition for each independent experiment) as described in [20].

### 2.8. In Vivo Experiments

Nude mice tumour xenografts were performed in compliance with the French guidelines for experimental animal studies (Direction des services vétérinaires, ministère de l’agriculture, agreement #1078) as described in [36,39]. Briefly, 2 × 10^6^ SW620 or Lovo cells (or derivatives) were subcutaneously injected in the flank of 5-week-old female athymic nude mice (Envigo). Tumour volumes were measured as the indicated intervals using callipers. After 20 days in the case of SW620 and 45 days in the case of Lovo cells, tumours were excised and weighed.

### 2.9. Statistical Analysis

All analyses were performed using GraphPad Prism (9.3.1) (graphpad software, San Diego, USA). Data are presented as the mean ± SEM from at least 3 independent experiments. When distribution was normal (assessed with the Shapiro Wilk test), the two-tailed *t*-test was used for between-group comparisons. In the other cases, the Mann-Whitney test was used. Statistical analyses were performed on a minimum of three independent experiments. The statistical significance level is illustrated with *p* values: * *p* ≤ 0.05, ** *p* ≤ 0.01, *** *p* ≤ 0.001 (*t*-test).

## 3. Results

### 3.1. PEAK2 Expression and Phosphorylation in CRC

We first evaluated PEAK2 expression and phosphorylation on its regulatory Y413 [10,21,27] in CRC from a panel of 11 CRC cell-lines. We found high PEAK2 expression and phosphorylation in advanced-stage CRC cell-lines (Figure 1A). Interestingly, Y413 PEAK2 phosphorylation correlated with SRC family kinases (SFKs) activity (Figure 1A), previously identified as an essential activation mechanism of PEAK1_3 signalling [22]. PEAK2 expression was also detected in patient-derived CRC lines. These include CPP 24, 43 and 44 cell lines that have been freshly established from primary tumour samples that express mutated KRAS, and the CTC44 and 45 cell lines derived from circulating tumour cells (CTC) [33,34]. Consistent with the essential role of CTCs in metastasis development, these CTC lines show self-renewal capacities and metastatic properties when injected in the spleen of nude mice [33,34]. PEAK2 phosphorylation was detected in CTCs (Figure 1B) which correlated with a high level of SFK activity (Figure 1B). We next examined the clinical relevance of our findings using CRC specimens. Transcriptomic analysis of 205 specimens of stage III CRC [35] revealed that high level of *PRAG1* expression, which encodes PEAK2, was associated with shorter relapse-free survival (RFS) (Figure 1C). Collectively, these results suggest that PEAK2 expression and its tyrosine phosphorylation may be associated cancer progression, implicating an oncogenic SFK-dependent mechanism.

### 3.2. PEAK2 Protumour Function in CRC

We next assessed PEAK2 role in CRC cell behaviour. For this, we depleted PEAK2 expression by a shRNA strategy in two KRAS mutated CRC cell-lines that expressed high endogenous PEAK2 level (i.e., SW620 and Lovo) (Figure 2A). We found that PEAK2 silencing reduced the elongated morphology of these tumour cells (Figure 2B) and cell invasive capacity when assayed in Boyden chamber coated with matrigel (Figure 2C). At the molecular level, PEAK2 depletion also affected tyrosine phosphorylation of specific proteins, consistent with phosphotyrosine-dependent PEAK2 signalling operating in these cells (Figure 2A). PEAK2 depletion also diminished anchorage-independent cell growth, as measured on number of colonies (Figure 3A and Appendix A). In vivo, subcutaneous injection of these tumour cells in *nude* mice produced similar results, i.e., 35–40% reduction of tumour development upon PEAK2 depletion, as evaluated on the volume and the mass of the tumours (Figure 3B,C). We next evaluated PEAK2 cancer activity by transducing a rat PEAK2 expression construct (named PEAK2^Myc^) in KRAS-mutated HCT116 CRC cells that expressed low endogenous PEAK2 (Figure 4A). PEAK2^Myc^ expression increased anchorage-independent growth, as shown on the number of colonies produced (Figure 4A). It also induced tyrosine phosphorylation of specific proteins including PEAK2^Myc^ [11,19,22] (Figure 4A). A structure-function analysis of PEAK2 revealed an important function of Tyr413 residue (PEAK2^Myc^ YF) unlike the predicted ATP binding site (PEAK2^Myc^ KD), demonstrating the pseudo-kinase nature of PEAK2 signalling (Figure 4A). We next evaluated PEAK2^Myc^ transforming capacity in SW620 cells by a similar method (Figure 4B); however PEAK2^Myc^ expression alone did not reveal any cellular growth or invasive effect (Figure 4B,C). Still, a promoting effect was noted on CRC cells adhesion on collagen matrix (Figure 4C). However, PEAK2^Myc^ tyrosine phosphorylation was not detected either (Figure 4B), suggesting that PEAK2 expression alone is poorly oncogenic in these cells. We next addressed whether forced PEAK2^Myc^ tyrosine phosphorylation would promote an oncogenic effect. Consistent with this, SRC coexpression induced PEAK2^Myc^ tyrosine phosphorylation (Figure 4B), which enabled PEAK2-dependent cell adhesion, invasion and colony formation (Figure 4B,C). Collectively, our results uncover a PEAK2 oncogenic activity in CRC cells, which implicates a tyrosine phosphorylation-dependent mechanism.

### 3.3. Phosphoproteomic Analysis of PEAK2 Signalling

We next searched for oncogenic pathways activated by PEAK2 in CRC cells. Using a candidate approach, we observed that PEAK2 depletion had no inhibitory effect on MAPK and AKT activities in SW620 and Lovo CRC cell-lines (Appendix A). PEAK2 depletion also did not affect SFK activity either, consistent with PEAK2 acting downstream SRC in CRC cells (Appendix A). Despite the fact that PEAK2 could regulate Notch and AMPK pathways [11,18], its depletion had no clear effect on these signalling pathways in CRC cells (Appendix A). We then turned to a global phosphoproteomic analysis centred on tyrosine phosphorylation because of the TK-dependent nature of PEAK2 signalling (Figure 5A). Such analysis was initiated by transient expression of PEAK2^Myc^ in HEK293T cells that produced a high level of cellular protein tyrosine phosphorylation [11]. Phosphotyrosine peptides were immune-purified from trypsin-digested cell lysates expressing or not PEAK2^Myc^ and analysed by a label-free mass spectrometry-based analysis using LC-MS/MS. From a total of 522 quantified phosphopeptides with a log_2_ fold change (FC) ≥1.5 upon PEAK2^Myc^ expression (Appendix A), we selected 30 tyrosine phosphorylation sites detected in 2 out 3 biological experiments (Figure 5B and Appendix A). We attributed the heterogeneous phosphotyrosine profiles by the difference in PEAK2^Myc^ expression levels from biological replicates (Appendix A). From this analysis, we identified several signalling proteins (e.g., PI3K, SHIP2, ABL, CSK, LYN, RACK1 and p38 MAPK) and regulators of F-actin assembly (e.g., Cortactin, Debrin-like protein, Cofilin, MRCK beta, delta-Catenin, Nck) and cell adhesion (e.g., Desmoplakin, AMOT, MAGI1, PAR3, Podoplanin 1) as PEAK2 targets (Figure 5B and Appendix A). These results were highly consistent with our previous interactomic analysis, which revealed CSK as the main associated TK and regulators of actin dynamics [11], such as the actin binding Arp2/3 complex, as PEAK2 interactors (Appendix A).

### 3.4. An Interplay between ABL and PEAK2 in Phosphotyrosine Signalling

The fact that PEAK2 induced phosphorylation of ABL on the activating Y204 residue was unexpected and placed ABL as a novel TK activated by this pseudo-kinase (Figure 5B). We then analysed this molecular response further by coexpressing PEAK2^Myc^ and ABL in HEK293T cells (Figure 5C). We found that PEAK2 induced a large increase in ABL kinase activity, as measured on Y412 phosphorylation, which is located in the kinase activation loop of ABL and is required for kinase activity (Figure 5C) [40]. ABL expression also induces Y413 PEAK2 phosphorylation, uncovering an interplay between ABL and PEAK2 tyrosine phosphorylation (Figure 5D). This notion was further supported by the reduction of pY413 PEAK2 level upon siRNA-mediated ABL silencing or ABL pharmacological inhibition (Figure 5D and Figure 6A). Similarly, ABL-like inhibitors (i.e., nilotinib, bosutinib and dasatinib) reduced PEAK2-induced protein tyrosine phosphorylation, known to be mediated by phosphorylation of Y413 PEAK2-(Figure 6A) [41]. Finally, ABL and PEAK2^Myc^ coexpression induced a remarkable increase in protein tyrosine phosphorylation. This activity was inhibited by pharmacological inhibition of ABL, but not CSK, thus excluding a major role for CSK in this activation process (Figure 6C). We thus concluded to a forward feedback activation loop between ABL and PEAK2 and identified ABL as a novel important component of PEAK2 phosphosignalling.

### 3.5. ABL Activity Regulates PEAK2-Induced Filopodia in CRC Cells

Finally, we aimed at validating these results functionally by focusing on PEAK2 signalling during tumour cell migration. Since PEAK2 interacts with regulators of F-actin cytoskeleton [11,42], we first checked whether PEAK2 colocalizes with F-actin cytoskeleton in CRC cells. SW480 cells stably expressing PEAK2^Myc^ were seeded on fibronectin and PEAK2 cellular localization was evaluated by indirect immunofluorescence using anti-Myc antibody. By this approach, PEAK2^Myc^ was readily detected at the cell periphery and more specifically at F-actin containing adhesion structures (Figure 7A). Interestingly, mutation of the main PEAK2 tyrosine phosphorylation site (YF) or the ATP binding site (KD) had no effect, highlighting the scaffolding and phosphotyrosine-independent nature of PEAK2 localization at focal adhesions (Figure 7A). Similar results were obtained in SW620 CRC and in SRC-transformed 3T3 cells (SrcYF-NIH3T3) (Appendix A). Additionally, PEAK2 localized at invadosomes of SRC-transformed cells (Appendix A) [43], consistent with a role for this pseudokinase in cell invasion.

We next searched for a PEAK2 function on cytoskeletal dynamics of CRC cells. Transient expression of PEAK2^Myc^ in HCT116 cells seeded on fibronectin induced a dramatic increase in fingerlike protrusions of actin filaments called filopodia (number and length) (Figure 7B,C). These protrusions play an important role in directed tumour cell migration, by sensing the chemical and physical environment [44]. A mutagenesis analysis revealed an essential role for Y413 PEAK2 on this cytoskeletal change (YF mutant), unlike the ATP binding site (KD mutant) (Figure 7B,C). In agreement, fibronectin also enhanced Y413-dependent PEAK2 phosphotyrosine signalling in HEK293T cells (Appendix A), thus confirming an interaction between PEAK2 and integrin signalling. ABL has been established as a central regulator of filopodia activity in various cell systems, including neuronal and transformed epithelial cells [45,46]. Consistent with our biochemical data showing an interplay between PEAK2 and ABL activity, this PEAK2 function was strongly inhibited by pharmacological ABL kinase inhibition (Figure 7C) [41]. We thus concluded to an important role of ABL in cytoskeletal dynamics induced by PEAK2 expression in CRC to enable tumour cell invasion.

## 4. Discussion

### 4.1. PEAK2 Tumour Activity in CRC

Here, we report a tumour function for the PEAK2 pseudokinase in CRC. PEAK2 overexpression facilitates growth and invasive behaviour of CRC cells in vitro while its depletion reduces these transforming properties. PEAK2 depletion also reduces tumour formation in nude mice. These results corroborate previous reports showing a PEAK2 proinvasive role in various cell lines derived from tumours of epithelial origin, including from pancreatic and gastric cancers [19,21]. Importantly, our functional results corroborated structural data [11,12,25] and establish a scaffolding role of PEAK2 in this malignant process. The clinical relevance of this PEAK2 activity was supported by correlation between PEAK2 transcript level and the shorter relapse free survival in a cohort of CRC patient samples. 

While *PRAG1* was found rarely mutated in CRC (www.cbioportal.org, accessed on 14 March 2022), PEAK2 oncogenic function may primarily involve aberrant protein expression. However, our results suggest that overexpression alone may not be sufficient to induce its transforming effect and that PEAK2 tyrosine phosphorylation is a key determinant of its transforming activity. Indeed, all PEAK2 transforming activities we observed in CRC cells were diminished upon inactive mutation of its main tyrosine phosphorylation site. Furthermore, PEAK2 overexpression harbouring low Y413 PEAK2 phosphorylation was poorly oncogenic, while the promotion of PEAK2 tyrosine phosphorylation increased its transforming effect. Interestingly, our results identified SRC as an important activator of this PEAK2 function. This idea was corroborated by our previous phosphoproteomic analysis in CRC cells who identified PEAK2 as an important SRC oncogenic substrate [22] and the strong correlation between the pY413 PEAK2 level and SRC activity in established and primary culture of patient-derived CRC cell lines. However, additional upstream TKs may be expected in this signalling process including EGFR, which is frequently overactivated in metastatic CRC [2,5].

### 4.2. PEAK2 Adhesive Activity in CRC Cells

PEAK2 was originally identified as regulator of cell adhesion and morphology; therefore, it is not surprising that its aberrant expression affects adhesive properties of CRC cells that facilitate cancer progression. This idea is corroborated by previous reports showing an important role for PEAKs in the induction of epithelial to mesenchyme transition of other cancer cells [19,21], and our results showing a diminution of the elongated morphology of CRC cells upon PEAK2 silencing. This notion is reinforced by the prominent localization of PEAK2^Myc^ at focal adhesions and its capacity to promote cytoskeletal dynamics of CRC cells. We thus propose that PEAK2 over-activation may affect adhesive capacity of CRC cells by inducing F-actin enriched membrane protrusions involved in cell invasion. For instance, PEAK2 overexpression induces filopodia formation, which may promote directed cancer cell migration during metastatic dissemination [44]. PEAK2 may also participate in invadosomes formation to enable extracellular membrane degradation [43]. Interestingly, our results also suggest a promoting role of PEAK2 in CRC growth. The underlying mechanism is unclear although one obvious mechanism would implicate its cell adhesive function. Interestingly, PEAK2 has been linked to Notch transcriptional activity during development and tumour formation [18]. However, we did not detect any clear PEAK2 nuclear localization, nor any effect of PEAK2 depletion on Notch signalling in CRC cells studied. Therefore, PEAK2 may use additional signalling mechanisms to induce CRC development.

### 4.3. PEAK2 Signalling in CRC Cells

Our report brings novel insight into PEAK2 signalling in CRC. While PEAK2 does not regulate major RTK pathways, e.g., MAPK or AKT signalling, its expression induces a phosphotyrosine signalling despite its catalytic inactivity. From proteomic analyses, we and others reported that PEAK2 can use the TK CSK to induce protein tyrosine phosphorylation [11,19]. Complex formation primarily involves SRC-induced Y413 PEAK2 phosphorylation with the SH2 domain of CSK [11,47]. Importantly, PEAK2 uses a SHED dimerization mechanism to induce CSK activation enabling phosphorylation of novel substrates [11]. Moreover, PEAK2 was identified as a new CSK substrate uncovering a feed-forward CSK activation loop that promotes cell motility [19]. Consistent with this, we observed that PEAK2 promoted CSK tyrosine phosphorylation on Tyr18 lying in its SH3 domain, which may affect CSK-SH3 binding capacity, enabling kinase dimerization and phosphosignalling [48,49]. While CSK displays antioncogenic activity in some cancers by inactivation of SFKs, this mechanism is impaired in CRC [37,50]. Furthermore, CSK was found strongly overexpressed in CRC along with aberrant expression of SFK activity and patients display CSK autoantibodies, which may define a novel biomarker of the disease [51]. CSK overexpression also increases growth and invasive properties of advanced CRC cells and induces migration of epithelial tumour cells in a PEAK2-dependent manner [19,37]. Conversely, all PEAK2 transforming activities we observed in CRC cells required Y413 phosphorylation, which defines the main CSK binding site [11,19,47]. Collectively, these observations suggests that CSK may contribute to PEAK2 oncogenic signalling in CRC. 

Our phosphoproteomic analyses identified additional components of PEAK2 signalling, including regulators of F-actin assembly, cell adhesion and polarity. This finding is highly consistent with the cell adhesive activity of PEAK2 from our study. Surprisingly, this analysis uncovered an additional TK involved in PEAK2 signalling. Like CSK, we detected an interplay between ABL and PEAK2, which creates a feedforward ABL activation loop, as revealed by the remarkable increase in cellular protein tyrosine phosphorylation upon PEAK2 and ABL expression. Our results also place PEAK2 as a novel ABL activator. Mechanistically, we propose that like CSK, dimeric PEAK2 would act as a scaffold protein to promote ABL kinase dimerization and activation. This model is supported by previous interatomic analyses who identified PEAK2 as an ABL interactor [52]. How PEAK2 interacts with ABL is unclear and this point deserves further investigation. The biological importance of this finding was next revealed by the ABL-dependent nature of PEAK2-induced F-actin assembly in CRC cells. Consistent with our results, several reports identified a pro-tumour function of ABL in CRC. For instance, an EphB-Abl signalling pathway has been associated with intestinal tumour initiation and growth [53], and ABL was shown to be an important component of the Notch invasive signalling cascade in CRC [54]. Overall, our results suggest that PEAK2 utilizes several TKs to mediate its tumour function and reveals a complex nature of pseudo-kinase signalling in cancer. 

## 5. Conclusions

Our results uncover an important tumour function of the pseudokinase PEAK2 in CRC development. PEAK2 transforming activity implicates deregulation of CRC cell adhesive and migratory properties. Interestingly, this PEAK2 transforming function was observed in KRAS mutated CRC cells, for which there is an unmet medical need [2]. Therefore, PEAK2 would be an interesting therapeutic target in these tumours. Consistently, our proteomic analyses uncovered several TKs, including ABL, activated by PEAK2 that would mediate its transforming activity, for which several inhibitors have been developed for the clinic (e.g., nilotinib, dasatinib and bosutinib). Our results predict inhibition of PEAK2 tumour activity by these therapeutic agents, which would also contribute to their antitumour effects reported in some CRC [50,55,56]. Collectively, our results suggest that PEAK2 pseudokinases could define attractive therapeutic targets in CRC, despite its catalytic inactivity.

## Figures and Tables

**Figure 1 cancers-14-02981-f001:**
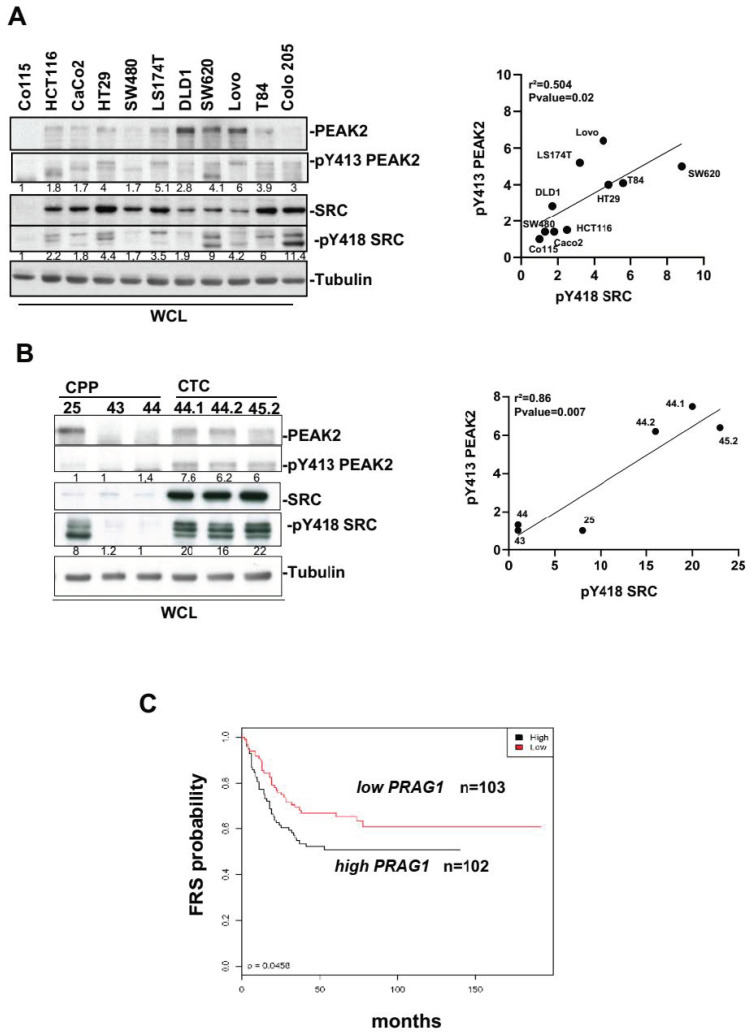
PEAK2 expression and tyrosine phosphorylation in CRC. (**A**,**B**): PEAK2 protein levels and Y413 phosphorylation in established CRC cell lines (**A**) and patient-derived CRC lines (**B**). Left panel: Western blotting showing the levels of PEAK2, pY413 PEAK2, SRC, and pY418 SRC from indicated CRC cell lines and CRC patient-derived lines. Right: correlation between the mean (*n* = 3) of relative pY413 PEAK2 level (fold control) and relative SKF activity (pY418 SRC level, fold control) from indicated CRC or CTC cells. (**C**) Patients with CRC showing high *PRAG1* expression have shorter free recurrence survival (FRS). Kaplan–Meier analysis using data from 205 patients with stage III CRC subdivided according to the tumour *PRAG1* expression level (high/low) (*p*-value: 0.0450).

**Figure 2 cancers-14-02981-f002:**
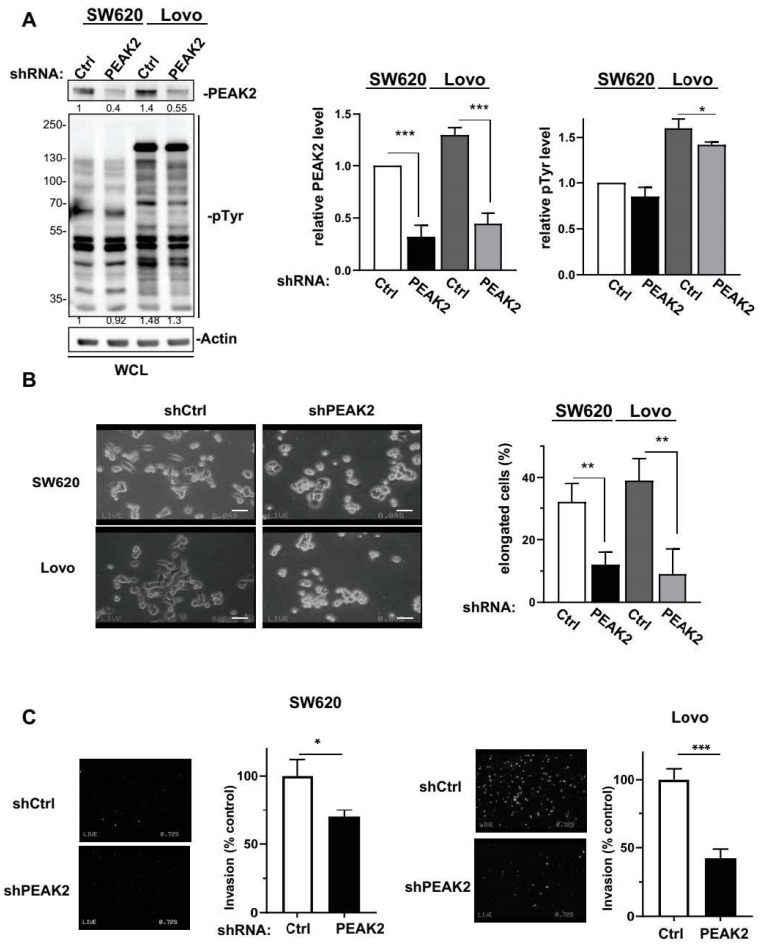
PEAK2 regulates CRC cell morphology and invasion. PEAK2 depletion by shRNA inhibits CRC cell growth and invasion. (**A**) Cellular protein tyrosine phosphorylation and PEAK2 level in CRC cells transduced with indicated shRNA (left: a representative example, right: quantification). (**B**) Morphology of CRC cells transduced with indicated shRNA (left: a representative example, right: quantification of the % of elongated cells). (**C**) Invasion of CRC cells transduced with indicated shRNA in Boyden chambers coated with matrigel (left: presentative image, right: quantification of number of invaded cells). Mean ± SEM; *n* = 3; * *p* < 0.05; ** *p* < 0.01; *** *p* < 0.001 (Student’s *t*-test).

**Figure 3 cancers-14-02981-f003:**
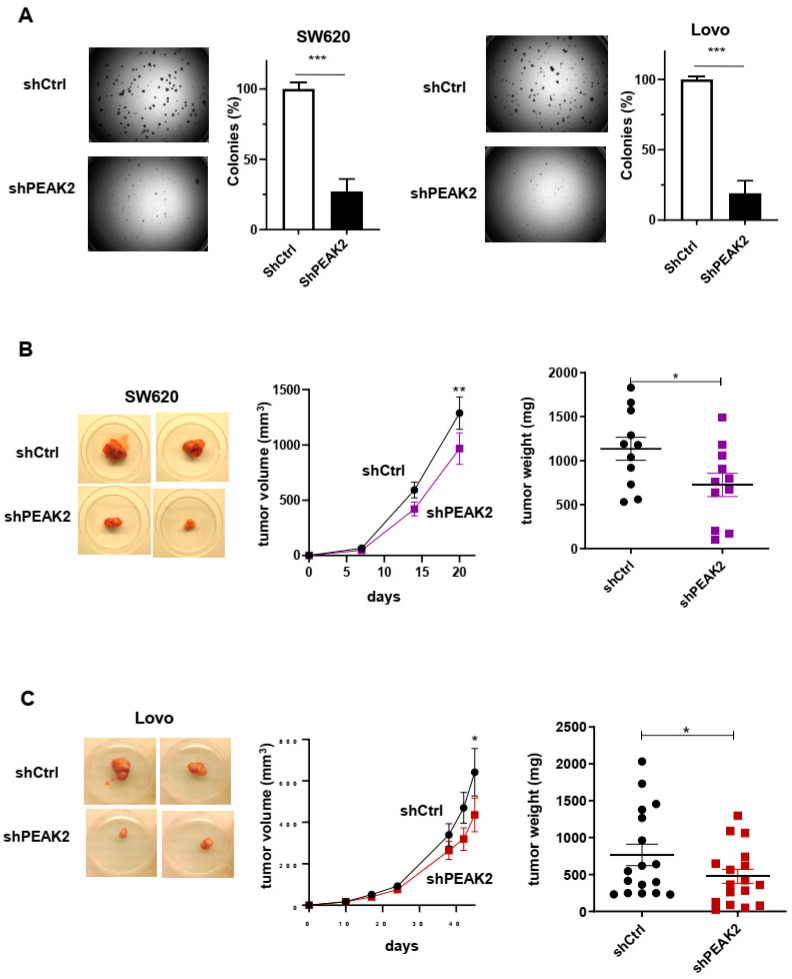
PEAK2 depletion reduces CRC cell growth and tumour development. (**A**) Anchorage-independent growth in soft agar of CRC cells expressing shRNA control (shCtrl) or shRNA PEAK2 (shPEAK2) (left: representative example; right: quantification of colonies obtained in soft-agar. (**B**,**C**) Tumour development in nude mice subcutaneously inoculated with indicated CRC cells that were transduced with shRNA control (shCtrl) or shRNA PEAK2 (shPEAK2). Is shown a representative example of tumours obtained in nude mice, the time-course of tumour development (volume) and the tumour mass; mean ± SEM from 13 (**B**) and 17 (**C**) mice per condition (* *p* < 0.05; ** *p* < 0.01; *** *p* < 0.001) (Student’s *t*-test).

**Figure 4 cancers-14-02981-f004:**
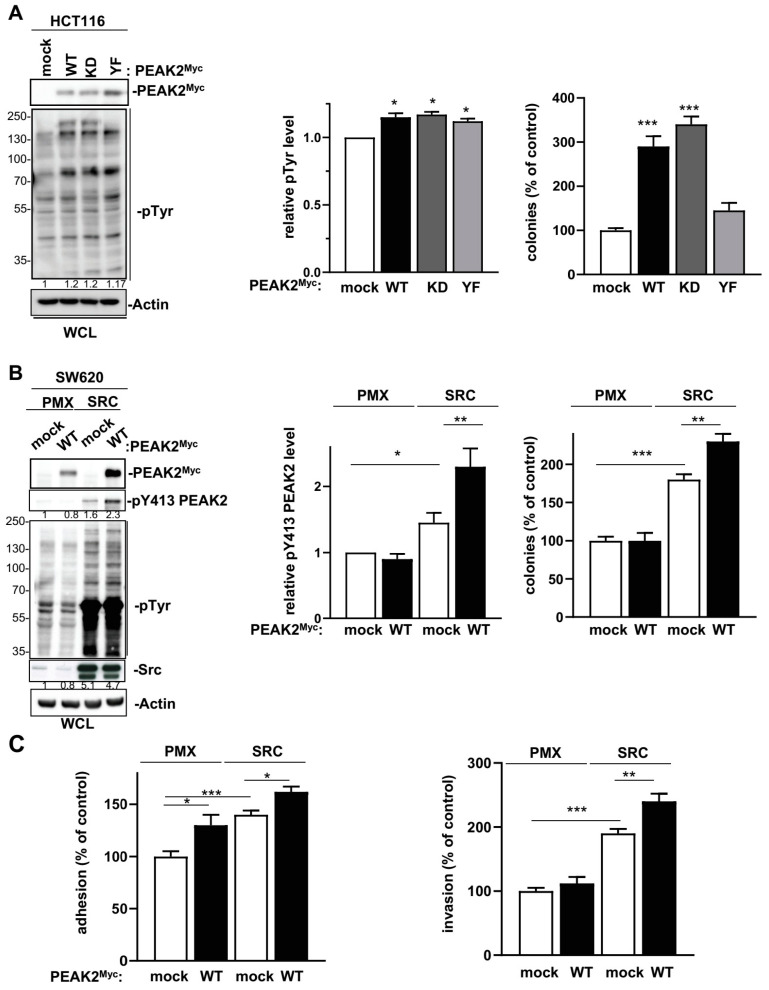
PEAK2 expression promotes CRC cell growth, adhesion and invasion. (**A**) PEAK2 expression increases growth of HCT116 CRC cells. Is shown the cellular protein tyrosine phosphorylation, PEAK2 protein levels and anchorage independent growth of HCT116 cells transduced with indicated PEAK2 constructs. The quantification of protein tyrosine phosphorylation and colonies formation in soft agar (% control) is shown from 3 independent experiments. (**B**) SRC expression promotes PEAK2 oncogenic function in SW620 CRC cells. Protein tyrosine phosphorylation, PEAK2 protein level and phosphorylation on Y413 and colonies formation in soft-agar of SW620 cells that were coinfected with SRC, PEAK2^Myc^ or control viruses (mock, PMX) as shown. The relative quantification of pY413 PEAK2 levels and colonies formation is shown from 3 independent experiments. (**C**) Adhesion on fibronectin and invasion in matrigel of indicated SW620 cells. Is shown the mean ± SEM; *n* = 3–4 * *p* < 0.05; ** *p* < 0.01; *** *p* < 0.001 (Student’s *t*-test).

**Figure 5 cancers-14-02981-f005:**
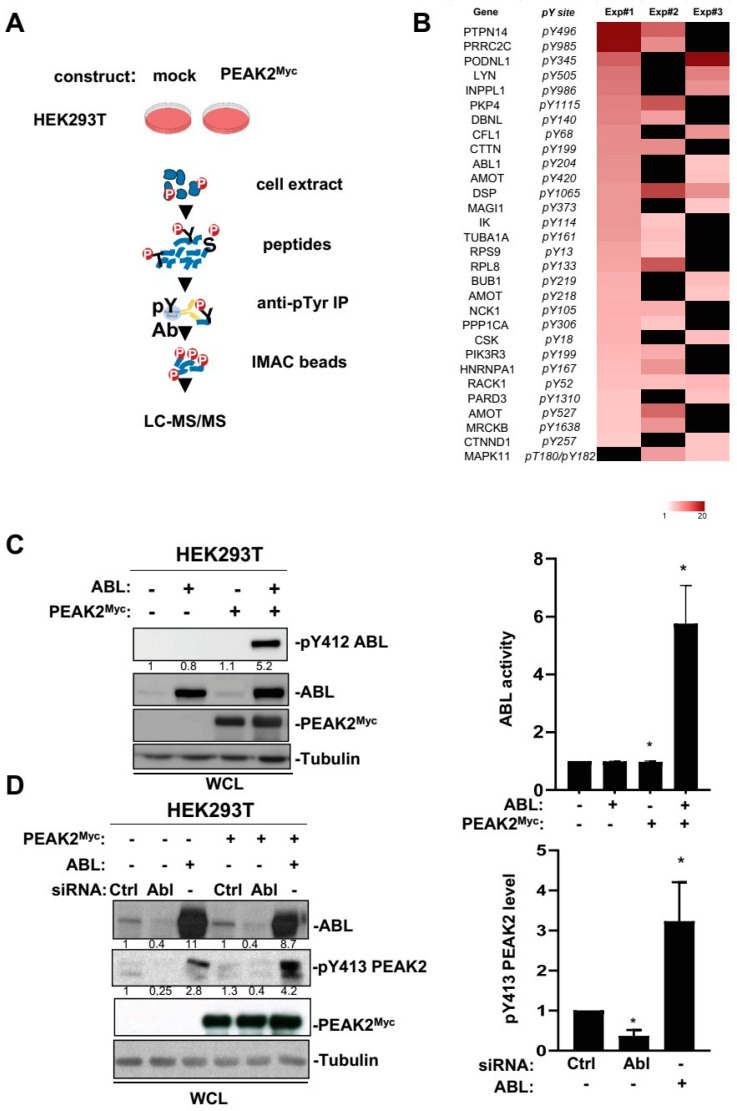
Phosphoproteomic analyses identified ABL as a novel PEAK2 target. (**A**) Workflow of the phosphoproteomic analysis of PEAK2 signalling in HEK293T cells. (**B**) Heatmap of phosphotyrosine peptides that show a log_2_ fold change (FC) ≥ 1.5 upon PEAK2^Myc^ expression in 2 out 3 biological independent experiments. Missing values are shown in black. (**C**) PEAK2 induces ABL tyrosine phosphorylation and activation. (**D**) ABL induces PEAK2 phosphorylation on Y413. The levels of PEAK2^Myc^, ABL, pY413 PEAK2, and pY412 ABL in HEK293T cells transfected with indicated reagents are shown. Right panels represent the relative quantification of indicated signals (mean ± SEM; *n* = 3–4; * *p* < 0.05; Student’s *t*-test).

**Figure 6 cancers-14-02981-f006:**
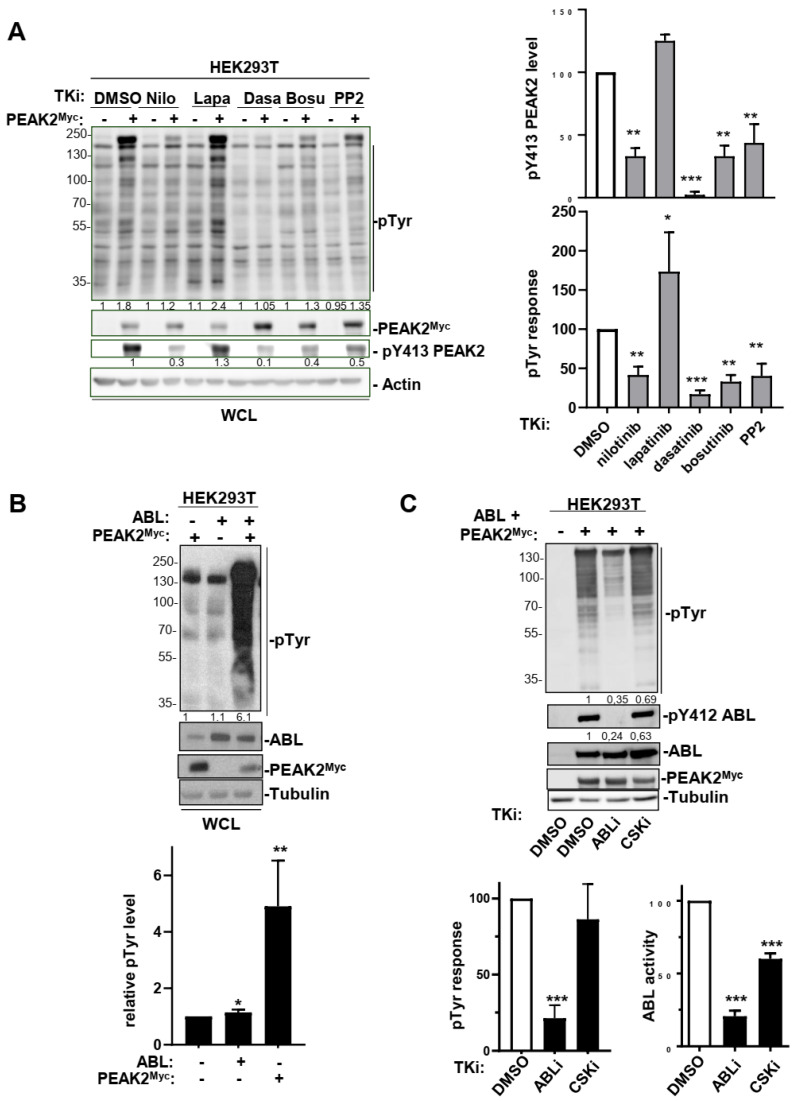
An interplay between ABL and PEAK2 activity. (**A**) Protein tyrosine phosphorylation induced by PEAK2 expression is diminished upon ABL pharmacological inhibition. Is shown the level of protein tyrosine phosphorylation of HEK293T cells transfected with PEAK2^Myc^ and treated for 2 h with indicated TK inhibitors (0.1% DMO; nilotinib, dasatinib and bosutinib: 100 nM; PP2 and lapatinib: 5 μM). The levels of PEAK2^Myc^, pY413 PEAK2 and protein tyrosine phosphorylation (pTyr). Right panels represent the relative quantification of indicated signals (mean ± SEM; *n* = 3; * *p* < 0.05; ** *p* < 0.01; *** *p* < 0.001 Student’s *t*-test). (**B**) ABL activity potentiates PEAK2^Myc^-induced protein tyrosine phosphorylation in HEK293T cells. (**C**) CSK does not play a major role in this activation process. Transfected cells were treated for 3 h with indicated TK inhibitors (ABLi nilotinib: 100 nM; CSKi: 100 nM). Bottom panels represent the relative quantification of indicated signals (mean ± SEM; *n* = 3; * *p* < 0.05; ** *p* < 0.01; *** *p* < 0.001 Student’s *t*-test).

**Figure 7 cancers-14-02981-f007:**
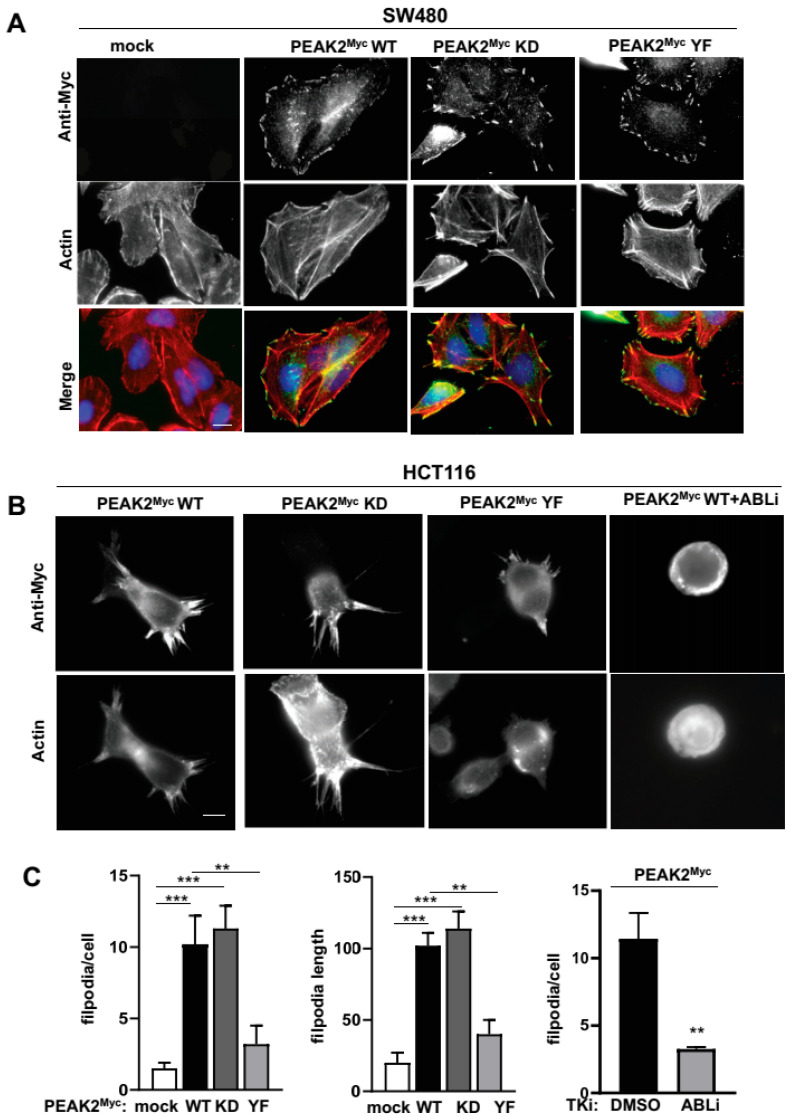
PEAK2 induces ABL-dependent folipodia in CRC cells. (**A**) PEAK2 colocalizes with F-actin structures at focal adhesion of CRC cells. Immunostaining of PEAK2 and F-actin in SW480 cells expressing indicated PEAK2^Myc^ constructs that were seeded on fibronectin. (**B**,**C**) PEAK2 induces ABL-dependent filopodia. A representative example (**B**) and quantification (**C**) of filopodia in HCT116 transfected with indicated PEAK2^Myc^ constructs and treated with DMSO (0.1%) or ABLi (100 nM nilotinib) as indicated for 2 h before cell fixation; mean ± SEM; *n* = 3; ** *p* < 0.01; *** *p* < 0.001 Student’s *t*-test).

## Data Availability

The data presented in this study are available in the article and in Appendix A.

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
