# Peer review of "Oncogenic Signalling of PEAK2 Pseudokinase in Colon Cancer"

_cancers, 2022, doi:10.3390/cancers14122981_

Round 1
Reviewer 1 Report
line 125 -> change caxo-2
Have you looked at the Y530 dephosphorylation of src kinase, to assess its activation?
fig2A -> Have you performed a quantification of p-Tyr (as you were able to do for figure 4A)? the western blot is not significant on its own.
Did you analyze the expression level of the Peak2 protein at the end of each in vivo experiment?
Have you carried out analyzes of the expression levels of the different integrins after transfection with shPeak2 and Peak2myc?
Why change adhesion matrix (collagen in figure 4C and fibronectin in figure 7 A and B)? The integrins involved for these different adhesion matrices are not the same, so the activated signaling pathways are not totally the same and moreover there is not the same action on the actin network.
How many cells were analyzed for the different immuno-fluoresence experiments?
The rest of the paper is well written and understandable. Thanks
Author Response
Referee #1
We thank the referee with their positive comment and suggestions.
Comments and Suggestions for Authors
line 125 -> change caxo-2
This point has been corrected accordingly.
Have you looked at the Y530 dephosphorylation of src kinase, to assess its activation?
We have not looked at pY530 SFK levels but pY418 SFK is a very good surrogate for SFK activity. Now, SFK activation can be achieved in the absence of Y530 dephosphorylation indicating that Y530 level is not the best marker for assessing SFK activity.
fig2A -> Have you performed a quantification of p-Tyr (as you were able to do for figure 4A)? the western blot is not significant on its own.
Quantification of p-Tyr and other WB signals (relative to tubulin/actin level) has been incorporated in all figures of the revised ms. There is indeed no statistical significant difference on pTyr signals from SW620, unlike Lovo cells. Nevertheless, we do see a diminution of specific tyrosine phosphorylation events from our WBs. This point is indicated in the results part (“At the molecular level, PEAK2 depletion also affected tyrosine phosphorylation of specific proteins, consistent with a phospho-tyrosine-dependent PEAK2 signalling operating in these cells”)
Did you analyze the expression level of the Peak2 protein at the end of each in vivo experiment?
We did not perform such analysis mainly because our anti-PEAK2 antibody does not produce a very good WB signal from tested tumor samples. This can be attributed, at least partially, to their cell heterogeneity. Besides, we keep these precious samples for future analyses of PEAK2 signaling in CRC. Now, one may predict a diminution of PEAK2 downregulation in some tumors, which may affect the level of tumor growth inhibition. Still, we detected a statistical significant diminution of tumor development upon PEAK2 depletion, consistent with our conclusion (i.e. a pro-tumoral function of PEAK2 in CRC).
Have you carried out analyzes of the expression levels of the different integrins after transfection with shPeak2 and Peak2myc?
We did not perform such experiment because the rationale behind this experiment is not evident, i.e. PEAK2 regulation of integration expression. For instance, we did not find any integrin in our PEAK2 interactomic analysis (Lecointre et al Structure 2018). Besides, we are not aware of any published reports showing any effect of PEAKs pseudo-kinases on integrin expression.
Why change adhesion matrix (collagen in figure 4C and fibronectin in figure 7 A and B)? The integrins involved for these different adhesion matrices are not the same, so the activated signaling pathways are not totally the same and moreover there is not the same action on the actin network.
We agree with the reviewer that these matrices may have a different impact on PEAK2 signaling. Now, tumor collagen plays important roles during CRC development (see our papers son this, e.g. Lafitte et al Frontiers Oncology 2019, Jeitany et al EMBO Mol Med 2018); therefore, collagen has some relevance when assessing CRC cell adhesion on ECM. The use of fibronectin in Figure 7 was dictated by its capacity to induce filopodia formation in PEAK2-transfected CRC cells, a cellular process known to be dependent upon ABL activity. Therefore, this setting was ideal to probe a role of ABL in a PEAK2 adhesive function. Consistent with this, we observed that fibronectin enhanced Y413-dependent PEAK2 phospho-tyrosine signaling also in HEK293T cells. This data further supports a link between integrins and PEAK2 signaling and is now incorporated in the revised ms (new Fig S5).
How many cells were analyzed for the different immuno-fluoresence experiments?
>50 transfected cells were analyzed per condition for each independent experiment (n>3). This point is specified in the methods part of the revised ms.
Reviewer 2 Report
In the current manuscript, Lecointre et al. investigated the oncogenic potential and signaling of the pseudo-kinase PEAK2 in colon cancer. Specifically, they found that PEAK2 can be phosphorylated at Tyr413 and this phospho-PEAK2 can drive colon cancer cells mobility, invasion, and tumor development. In addition, they discovered that the ABL kinase a novel activating partner of PEAK2 and they form a positive feedback-loop to co-activate each other. Overall, the manuscript is well-written, and the experimental design is logical. The data presented by the authors to support their claims are clear and convincing. However, some experiments lack important controls, and some conclusions are not entirely justified. Below are my comments for the study.
1. Why did the authors use a rat PEAK2Myc? Is there no human PEAK2 available or is it not possible to clone human PEAK2 into a plasmid? Since rat and human PEAK2 have slightly different amino acid numbers, this makes the findings in the paper slightly less significant.
2. In Fig. 2 A, the difference in pTyr is only slightly visible for Lovo cells while there is completely no difference in pTyr for SW620 cells.
3. In Fig. S2, the authors knock-downed PEAK2 by shRNA and measured the protein levels of several signaling pathways. They wrote in the text that pAKT did not change upon PEAK2 knock-down, but when one looks at the western blot, there is a big increase in pAKT bands with shPEAK2. Furthermore, the quantification below the bands indicates “1”, “1.2”, “1”, “1.1”, but the visual difference between the bands looks much bigger than 20% and 10% respectively. This data seems to suggest that PEAK2 knock-down actually induced AKT phosphorylation.
4. For Fig. 3 B and C, it would be interesting to measure the pTyr protein level in the shCtrl and shPEAK2 xenograft tumors collected from mice.
5. For Fig. 4 B, the authors overexpressed SRC to drive PEAK2 phosphorylation and activation. However, there is no western blot confirmation of SRC overexpression in these samples.
6. In Fig. 5, the authors performed phospho-proteomics to identify phospho-proteins regulated by PEAK2 using HEK293T as a model system. Did they confirm the increase in pTyr in the three triplicate biological samples before proceeding with the big experiment?
7. It would be very interesting to perform Co-IP to see if PEAK2 can directly bind to ABL and/or SRC.
8. For Fig. 7 B, the authors did not include an important control which is Mock transfection with empty vector to compare the morphology the control and PEAK2-overexpressing cells.
9. Also, for Fig. 7B, it would be important to include a SRC inhibitor as well since the authors showed in other Figures that SRC is important for PEAK2 phosphorylation at Y413, and this phosphorylation event is crucial for the formation of filopodia as demonstrated by PEAK2Myc YF mutant.
Author Response
Referee #2
Comments and Suggestions for Authors
In the current manuscript, Lecointre et al. investigated the oncogenic potential and signaling of the pseudo-kinase PEAK2 in colon cancer. Specifically, they found that PEAK2 can be phosphorylated at Tyr413 and this phospho-PEAK2 can drive colon cancer cells mobility, invasion, and tumor development. In addition, they discovered that the ABL kinase a novel activating partner of PEAK2 and they form a positive feedback-loop to co-activate each other. Overall, the manuscript is well-written, and the experimental design is logical. The data presented by the authors to support their claims are clear and convincing. However, some experiments lack important controls, and some conclusions are not entirely justified. Below are my comments for the study.
We thank the referee with their positive comment and suggestions.
- Why did the authors use a rat PEAK2Myc? Is there no human PEAK2 available or is it not possible to clone human PEAK2 into a plasmid? Since rat and human PEAK2 have slightly different amino acid numbers, this makes the findings in the paper slightly less significant.
When we started this project, there was no human PEAK2 construct available. We therefore generated our PEAK2 reagents (antibody, purified proteins, expression constructs) from the rat sequence. Indeed, there is some slightly different amino acid number with the human form but the modular structure is conserved: crystallographic studies from Daly’s lab (Patel et al Nat Commun 2017) and our group (Lecointre et al, Structure 2018) revealed conserved features of the C-terminus and the unstructured N-terminus displays conserved interaction motives (e.g. SH3 and SH2 binding motives). Therefore, we are quite confident that our data obtained in rat PEAK2-myc overexpressing models is relevant to the human counterpart.
- In Fig. 2 A, the difference in pTyr is only slightly visible for Lovo cells while there is completely no difference in pTyr for SW620 cells.
Quantification of p-Tyr and other WB signals (relative to tubulin/actin level) is now incorporated in all figures of our revised ms. There is indeed no statistical significant difference on pTyr signals from SW620 cells, unlike Lovo cells. Nevertheless, we do see a diminution of specific tyrosine phosphorylation events from our WBs. This point is clearly indicated in the results part (“At the molecular level, PEAK2 depletion also affected tyrosine phosphorylation of specific proteins, consistent with a phospho-tyrosine-dependent PEAK2 signalling operating in these cells”)
- In Fig. S2, the authors knock-downed PEAK2 by shRNA and measured the protein levels of several signaling pathways. They wrote in the text that pAKT did not change upon PEAK2 knock-down, but when one looks at the western blot, there is a big increase in pAKT bands with shPEAK2. Furthermore, the quantification below the bands indicates “1”, “1.2”, “1”, “1.1”, but the visual difference between the bands looks much bigger than 20% and 10% respectively. This data seems to suggest that PEAK2 knock-down actually induced AKT phosphorylation.
We agree with the reviewer that we detect an increase in pAkt level (relative to the loading control), which may be higher when compared to Akt protein levels. This data seems to suggest that PEAK2 knock-down induces both Akt protein diminution and increased phosphorylation. However, we did not pursue this idea further. Based on the reviewer’s comment, we have modified the text in the revised ms accordingly by saying that “we observed that PEAK2 depletion had no inhibitory effect on MAPK and AKT activities”
- For Fig. 3 B and C, it would be interesting to measure the pTyr protein level in the shCtrl and shPEAK2 xenograft tumors collected from mice.
We did not perform such analysis mainly because we keep these precious samples for future analyses of PEAK2 signaling in CRC. Now, the interpretation of pTyr WB may be difficult because of the cell heterogeneity contained in the tumor samples.
- For Fig. 4 B, the authors overexpressed SRC to drive PEAK2 phosphorylation and activation. However, there is no western blot confirmation of SRC overexpression in these samples.
This omission is now corrected in the revised Fig 4B.
- In Fig. 5, the authors performed phospho-proteomics to identify phospho-proteins regulated by PEAK2 using HEK293T as a model system. Did they confirm the increase in pTyr in the three triplicate biological samples before proceeding with the big experiment?
WBs on the increase in pTyr from the three triplicate biological samples have been incorporated in the revised Fig S2B.
- It would be very interesting to perform Co-IP to see if PEAK2 can directly bind to ABL and/or SRC.
We have already performed co-ip experiments and we do see a SH2/SH3-dependent Src-PEAK2 association (not shown). Besides, we are currently deciphering the interplay between ABL and PEAK2; however we feel that this point is beyond the scope of this study.
- For Fig. 7 B, the authors did not include an important control which is Mock transfection with empty vector to compare the morphology the control and PEAK2-overexpressing cells.
Quantification of mock-transfected cells was already incorporated in the graphs from Fig 7B; however, we added a representative image in the revised figure.
- Also, for Fig. 7B, it would be important to include a SRC inhibitor as well since the authors showed in other Figures that SRC is important for PEAK2 phosphorylation at Y413, and this phosphorylation event is crucial for the formation of filopodia as demonstrated by PEAK2Myc YF mutant.
We agree with the reviewer that it would be interesting to address the effect of a SRCi on this PEAK2 cellular response. We however did not perform such experiment because of a matter of time (10 days for revision). Now, our work demonstrates an essential role of Y413 in this PEAK2 cellular response. It also reveals that Y413 phosphorylation can be induced by many TK, including ABL, CSK (which is overexpressed in CRC cells, see Sirvent et al Oncogene 2011) and SRC (Leroy et al Cancer Res 2010). To consolidate this idea and the interaction between fibronectin and PEAK2 signaling, we added data showing that fibronectin enhances Y413-dependent PEAK2 phospho-signaling in HEK293T cells (new Fig S5B). Because SRC is essential component of integrin signaling, it is likely that SRC participates in this PEAK2 phospho-signaling pathway. Finally, we previously published that SRC invasive activity is mediated by PEAK2 Y413 phosphorylation (Leroy et al Can res 2010), further supporting a SRC-PEAK2-ABL signaling inducing filopodia formation in CRC cells.
Round 2
Reviewer 2 Report
The authors have adequately addressed all my concerns. The current manuscript can be accepted for publication with a minor change.
Original Reviewer Comment: For Fig. 4 B, the authors overexpressed SRC to drive PEAK2 phosphorylation and activation. However, there is no western blot confirmation of SRC overexpression in these samples.
Authors' response: This omission is now corrected in the revised Fig 4B.
New Reviewer Comment: The revised Fig.4 B still does not have the Src protein level. I looked at the raw western blot file and I saw a Src protein blot there. Perhaps the authors mistakenly did not update their Fig. 4B after they performed the Src western blot.
Author Response
The revised Fig4B was included in the zip file containing the revised text of the ms in track-change mode + a set of the revised figures in a pdf format.
Best
Serge Roche